# The Structure of Temperament in Caregivers of Patients with Schizophrenia

**DOI:** 10.3390/ijerph20032119

**Published:** 2023-01-24

**Authors:** Kasper Sipowicz, Kamila Łuczyńska, Bartłomiej Bąk, Kacper Deska, Katarzyna Nowakowska-Domagała, Tadeusz Pietras, Dorota Podgórska-Jachnik, Elżbieta Małujło-Balcerska, Marcin Kosmalski

**Affiliations:** 1Department of Interdisciplinary Disability Studies, The Maria Grzegorzewska University in Warsaw, 02-353 Warsaw, Poland; 2The Second Department of Psychiatry, Institute of Psychiatry and Neurology in Warsaw, 02-957 Warsaw, Poland; 3Department of Clinical Pharmacology, Medical University of Lodz, 90-153 Lodz, Poland; 4Department of Clinical Psychology and Psychopathology, Institute of Psychology, Faculty of Educational Sciences, University of Lodz, 90-128 Lodz, Poland; 5Department of Educational Research, University of Lodz, 90-136 Lodz, Poland; 6Department of Pneumonology, Medical University of Lodz, 90-153 Lodz, Poland

**Keywords:** caregivers, temperament, schizophrenia, outpatient care, young adults

## Abstract

The onset of schizophrenia symptoms usually occurs in early youth. As a result, the parents of these patients usually become their caregivers. The role of a caregiver for a person with schizophrenia is a considerable mental and physical burden. Therefore, an interesting issue is what motivates these people to take up this challenge. It is probable that, apart from the moral imperative or kinship, the factor determining this decision is the personality structure of the caregiver. The aim of our study was to compare the structure of temperament (according to the model of temperament as formal characteristics of behavior developed by Jan Strelau) in caregivers of young adults (age 18–25 years) with schizophrenia with the structure of temperament of parents of healthy young adults still living in the family home under their care. The study group consisted of 64 people (51 women), who were taking care of young adults (aged 18–25 years) with schizophrenia, while the control group (53 people, 42 women) consisted of parents of healthy adults still living in the family home. Both groups were asked to complete a questionnaire of the authors’ own design on their demographic data as well as The Formal Characteristics of Behavior—Temperament Inventory to assess the temperament traits. The results were given in the number of points obtained on average in each dimension. Both groups did not differ in terms of size and age, with women predominating. Caregivers of young adults with schizophrenia had higher values of briskness (43.22 ± 4.45 vs. 42.90 ± 3.98, *p* = 0.032), emotional reactivity (46.02 ± 4.39 vs. 41.01 ± 3.12, *p* = 0.012) and activity level (44.01.89 ± 4.15 vs. 37.59 ± 4.77, *p* = 0.022) compared to the control group. The remaining dimensions of temperament: perseverance, sensory sensitivity, rhythmicity, and endurance did not differentiate between the two groups. The temperament structure of caregivers of young people with schizophrenia differs from the temperament structure of caregivers of healthy adults. Caregivers of sick people have higher values of briskness, emotional reactivity, and activity level compared to the control group.

## 1. Introduction

Family members play a key role in providing emotional support to young-adult schizophrenic patients [1,2]. They often provide financial resources to their charges as well. The moment when they are faced with their child’s diagnosis for the first time is a crisis moment for the caregivers [3], who require emotional support then. The crisis usually subsides and gives way to sadness, daily struggle with the child’s illness, elements of burnout and resignation, adaptive, psychosomatic, anxiety, and mood disorders, which develop over time [4,5].

Schizophrenia most often affects young people who have not started a family yet, hence both their parents, their mothers, or, less frequently, fathers usually become their guardians [6]. Generally, however, one of the parents is more involved in providing the care and it is usually the mother. The child’s illness significantly reduces the quality of life of the caregivers, is a psychological, physical, and economic burden, and can be, as mentioned, the cause of adaptive syndrome complicated by mood disorders and anxiety disorders [7]. In psychiatry, the term “burden” of the family members of people with schizophrenia has been introduced [8]. It is understood as the impact of the patient’s illness on the health and functioning of the family, or as a result of the impact of mental illness on people living together with the sick person. This concept was introduced and developed in the sixties of the twentieth century by Grad and Sainsbury [9,10,11] and has entered permanently into systemic family therapy and psychiatry [12,13,14]. The burden of schizophrenia is also the cause of stigmatization of the patients’ families [15]. One of the determinants of a family member taking on the role of a caregiver is not only kinship or moral imperative, but also probably the personality structure of the caregiver [16]. Personality is formed on the basis of temperament, conditioned biologically to a large extent, as well as under the influence of factors of the social environment in which the individual lives [17]. Temperament, according to the Theory of Temperament as a Formal Characteristic of Behavior developed by Strelau, determines how a person performs certain activities (e.g., accurately, quickly, cyclically, briskly), but does not deal with the content of these activities [18]. Strelau assumed in his concept that temperament is a formal characteristic of human behavior, i.e., a description of how a person performs activities. The factor analysis showed that the structure of temperament consists of the following dimensions: briskness, perseverance, rhythmicity, sensory sensitivity, endurance, emotional reactivity, and activity [19]. Briskness is a tendency to react quickly and maintain a high pace of activities. Perseverance means a tendency to maintain and repeat a behavior despite the cessation of the stimulus that triggered such behavior. Rhythmicity is the tendency to rhythmically perform activities related to aspects of the circadian rhythm of a person. Briskness, perseverance, and rhythmicity are temporal characteristics of behavior. Sensory sensitivity is understood as the ability to respond to low-intensity sensory stimuli and the tendency to detect small differences in stimulus intensity. Endurance is understood as the ability to adequately respond in situations requiring long-term and highly stimulating activity, manifested in resistance to fatigue and distractions. Emotional reactivity is a tendency to react intensely to emotional stimuli, manifested in high sensitivity and low emotional resilience. Activity, in turn, is a tendency to undertake a highly stimulating activity or to engage in behaviors that provide strong external stimulation from the environment. Sensory sensitivity, endurance, emotional reactivity, and activity represent the energetic level of human activity [18,19].

Temperament has a largely biological background, it depends on the properties of the central nervous system of the subject, neurodevelopmental and determined genetically [18]. The structure of temperament can, at least in theory, influence the choice of the social role of the direct caregiver of the sick person. On the other hand, the presence of schizophrenia in the family may have a modulating effect on the functioning of the central nervous system in the caregivers themselves. The question of the relationship between the structure of temperament and the choice to be a caregiver requires a thorough and careful analysis of the results of empirical research. It is interesting, however, that schizophrenic patients were more likely to have mutual antipathy with their caregivers than in non-clinical groups, which contradicts the results of our research discussed below [20].

In the PubMed database, there are no works devoted to the structure of temperament of caregivers of people with schizophrenia. There are also no publications on the personality structure of these caregivers.

## 2. Materials and Methods

### 2.1. Study Population

People who gave their written consent to participate in the study were included. The criterion for inclusion in the study group was caring for young adults in age 18–25 with schizophrenia and living together with them. These caregivers were randomly selected from a population of 120 patients treated in the District Psychiatric Outpatient Clinics in Pabianice and Aleksandrów Łódzki, in the Society of Friends of the Disabled in Łódź and the psychiatric outpatient department at the Institute of Psychiatry and Neurology in Warsaw. It should be emphasized that the main reasons for the refusal to participate in the study were the lack of declared willingness to cooperate with the doctor, negative attitude to pharmacological treatment, and extreme disapproval of the young adult’s illness. The control group consisted of adults caring for their young adult children aged 18–25 who did not become independent, and without mental disorders. Parents of such people were recruited from the pulmonary clinic and diabetes clinic of the N. Barlicki Teaching Hospital of the Medical University of Łódź. The study was conducted between 1 January 2019 and 31 January 2019 before the outbreak of the pandemic caused by the SARS-CoV-2 virus. The control group was deliberately selected so that the percentage of women and men was similar to that observed in the study group. The exclusion criteria for both groups were the age of the patient under care over 25 and under 18 years and lack of written consent to participate in the study.

### 2.2. Methods

Demographic data were obtained on the basis of a self-designed questionnaire containing information about the age and sex of caregivers, the person who is under the care, and other residents staying in the same house. The Formal Characteristics of Behavior—Temperament Inventory (FCB-TI) was used to assess temperament [19]. This questionnaire includes 7 dimensions orthogonal to each other and are briskness, perseverance, rhythmicity, sensor sensitivity, emotional reactivity, endurance, and activity. The value of individual dimensions was calculated on the basis of the sum of the points obtained (raw scores). The questionnaire is an original Polish tool built on the concepts of Jan Strelau. This concept is known in the world but rarely used in clinical trials. The study was approved by the Medical University of Lodz Bioethics Committee (approval document No. RNN/133/09/KE as amended).

### 2.3. Statistical Analysis

The normality of the distribution of the examined traits was assessed using the Shapiro-Wilk test. Means, variances and standard deviations, and correlation coefficients were calculated for the analyzed variables. As the distribution of variables was not in accordance with normality determined by the Shapiro–Wilk test, the differences between the 2 groups were assessed by the Mann–Whitney U test. A licensed StatSoft package (StatSoft GmbH: Hamburg, Germany), purchased by the Medical University of Lodz, was used for the calculations. *p* < 0.05 was considered statistically significant.

## 3. Results

Finally, the study group consisted of 64 caregivers (including 51 women), and the control group was 53 people (including 41 women). Both groups did not differ significantly in terms of the number of people included in the study (*p* = 0.39). The study involved mainly parents, in one case it was the sister, and in the other case the brother of the affected person. Therefore, the group other than the parents was not representative and was not taken into account in further analysis.

The demographic characteristics of the study groups are summarized in Table 1. No statistically significant difference in the mean age of people from both study groups was observed (*p* = 0.59). The percentage distribution of men and women in both groups was similar. Women accounted for approximately 80% of people providing direct care to patients with schizophrenia. Interestingly, there was no difference in the minimum and maximum ages of caregivers between both groups.

The results obtained are summarized in Table 2. The caregivers of young adults with schizophrenia had higher values of briskness (43.22 ± 4.45 vs. 42.90 ± 3.98, *p* = 0.032), emotional reactivity (46.02 ± 4.39 vs. 41.01 ± 3.12, *p* = 0.012) and activity (44.01.89 ± 4.15 vs. 37.59 ± 4.77, *p* = 0.022) compared to the control group of caregivers of healthy young adults. The remaining dimensions of temperament: perseverance (42.50 ± 3.29 vs. 43.10 ± 4.37), sensory sensitivity (42.36 ± 3.31 vs. 43.01 ± 3.29), rhythmicity (22.46 ± 4.15 vs. 23.01 ± 3.29) and endurance (36.46 ± 4.68 vs. 36.42 ± 5.52) did not differentiate between the two groups.

## 4. Discussion

In the group of caregivers of schizophrenia patients, briskness turned out to be higher than in the control group. The tendency to react quickly and maintain a high rate of activity is necessary at the time of exacerbation of schizophrenia in a child. Rapid response to symptoms of psychosis, or to the presence of suicidal ideation is one of the prerequisites for the good functioning of the patient [21]. This skill is usually trained by doctors and psychologists during the process of psychoeducation of caregivers and families of people with schizophrenia [22,23]. It is also reinforced in the process of systemic therapy of the families of schizophrenia patients [24,25]. Maintaining a high pace of the performed activities is useful during the patient’s health crisis. Taking the patient to an outpatient clinic or hospital, administering an increased dose of drugs or new drugs, vigilance to behavior, and monitoring suicidal and homicidal ideation requires intensive activity [26,27]. Therefore, the high intensity of the briskness dimension is a beneficial feature in caregivers of people with schizophrenia [28].

Emotional reactivity was also found to be greater in the group of people caring for an adult child with schizophrenia than in the control group. The direct caregivers of patients are usually women—their mothers. As shown in our research, fathers rarely take a dominant role in care. Women are more focused on reading emotions, especially those who care for children with special needs. Kulhara et al. suggest that the caregivers of patients with schizophrenia and psychotic disorders experience caring benefits (in the form of greater sensitivity to people with disabilities, greater emotional sensitivity, clarity concerning their life priorities, and a greater sense of inner strength), experience good aspects of their relationship with the patient, have positive personal experiences [29]. Some studies suggest that people whose experiences related to care are predominantly negative also have positive experiences related to this care in parallel, and this has to do with mindfulness of emotions [30]. It is interesting whether emotional reactivity correlates with a positive image of caring for a person with schizophrenia. Such studies would explain the sense of increased emotional reactivity in caregivers of people with schizophrenia. The publications cited above and the results of our research contradict the commonly accepted thesis formulated by Friede Fromm-Reichman that a schizophrenogenic mother is cold, unaffectionate, emotionally insensitive, issuing messages of a double bond character [31]. The thesis of a schizophrenogenic, emotionally insensitive mother became one of the leading hypotheses of systemic family therapy, contributing to the development of systemic therapy, although it largely blamed the child’s parents for the child’s illness [32]. Modern family therapy, however, has moved away from blaming parents (mothers) for the occurrence of schizophrenia in the family in favor of helping this family and their psychoeducation as well as providing them with support in the hardships of caring for the sick person [33,34].

Also, activity was observed to be greater in those caring for adult children with schizophrenia than in the control group. Activity is the search for behaviors that provide high external stimulation from the environment. It is partly related to Zuckerman’s construct of sensation-seeking trait [35,36]. A chronic illness of the child is a strong external stimulator. Perhaps the role of a direct caregiver in the family is taken by persons who need stimulation. On the other hand, caregivers of people with persistent psychotic symptoms often feel overwhelmed, stressed, exhausted, burdened, frustrated, or angry. The financial impact of care includes medical costs for the care recipients, the provision of financial support, and loss of productivity and income. Depression and anxiety are the common health effects for caregivers, who are also more likely to use physical health care resources compared to healthy controls [37]. High mental and physical activity in the caregivers of people with dementia has been demonstrated in the research to prevent the development of depression, anxiety, and burnout, and to increase life satisfaction [38]. It can be cautiously assumed that the empirically observed regularity also applies to the caregivers of people with other chronic diseases, including patients with schizophrenia. It has been shown that the most important predictor of overstrain is the nature of the caregiver-patient relationship in the case of acute schizophrenia. Additional predictors of burnout are the time spent with the patient and the burden resulting from limited social life and organization of free time, but not aggression or substance abuse [39]. We believe that high activity and demand for stimulation protect, at least partially, against excessive strain on the caregiver and against burnout.

The other dimensions of temperament did not differ between caregivers of people with schizophrenia and the control group. Our research indirectly shows that only those variables that differentiate both groups have an impact on taking up the role of a caregiver. Other temperamental variables probably do not affect the willingness to care for sick relatives. The converted standard nine (stanine) score values from the test normalization sample for the Polish population indicate that the values of these traits in both groups were within the range of average values calculated for the Polish population (stanine score 5) [19]. The values obtained for the dimensions that differentiated the studied groups in the control group were also within the range of stanine score 5, i.e., the value obtained for the Polish population, with the exception of briskness, which in the control group reached stanine score 4, a result slightly lower than the value obtained in the tested tool normalization sample. Briskness is a trait that decreases with age [19]. The population we studied included people with adult children, i.e., not very young. The normalization sample of the Polish population included people aged 18–65 [19].

### Limitations of the Study

The group of parents surveyed is relatively small, although large enough to draw statistical conclusions. The study should be repeated on a larger group with random control. It should be carried out on a multicenter basis. Additionally, the control group could be relatives of people with schizophrenia who did not decide to take on the role of caregiver. Unfortunately, in our opinion, it is impossible to gather such a group of doctors working with schizophrenic patients on a daily basis, because such people do not report to mental health clinics. People who do not provide care do not come with a sick person to the doctor for appointments.

It is not known whether the observed differences in the structure of temperament between caregivers of schizophrenia patients and people living with adult children result from the impact of the disease in children on the structure of the parent’s temperament, or whether the role of a caregiver is taken by people with a temperament structure appropriate for providing care. These two alternative hypotheses require in-depth longitudinal prospective studies and in-depth qualitative research on the lifestyle experiences of families with schizophrenia and on giving meaning to these experiences. The question arises whether the temperamental variable is not modified under the influence of chronic stress, which is caring for a person with schizophrenia. It is also necessary to consider how long-term stress affects emotional reactivity, and how overburdening chronic stress obscures innate temperamental traits, permanently modifying them. In Poland, research on the correlation between traumatic stress and temperament understood according to the concept of Strelau and his team has been conducted [40,41]. It can be cautiously assumed that caregivers of people with schizophrenia are exposed to chronic intense stress, but without the traumatic features causing post-traumatic stress disorder (PTSD) [42]. The effect of such stress on the structure of temperament requires further research.

It is also not known whether the observed temperament structure in caregivers of people with schizophrenia is not due to the common genetic heritage of predisposition to schizophrenia. A significant part of the variance of the trait of schizophrenia is known to be determined genetically [43]. It is also known that some neuropsychological deficits, which may also be a determinant of the formation of a specific temperament structure, are observed in healthy close relatives of people with schizophrenia [44]. This question should be considered in the broader context of temperamental predispositions to developing schizophrenia, if any [45]. No studies of the correlation between the structure of temperament consistent with Strelau’s Temperament Theory as a formal characteristic of behavior and the predisposition to develop schizophrenia have been conducted, which, in our opinion, should be done to verify the clinical usefulness of the model used.

Our work does not compare the results obtained in our study with the results of studies concerning the caregivers of people with other chronic diseases. We do not know whether the differences in temperament structure observed between healthy caregivers and caregivers of people with schizophrenia are specific to schizophrenia in the family, or are more universal for all chronic diseases that disrupt the functioning of the nuclear family. Single published papers reporting research on the temperament of caregivers of people with various diseases have never used the temperament theory model as a formal characterization of behavior and have typically used the concepts of Cloninger, Eysenck, or Akiskal’s affective temperament paradigm [46,47]. Therefore, we do not have any research to refer to.

The selection of the control group may raise objections not only due to the fact that the selection of the examined people is not random. People aged 20–25 who live with their parents are in some cultures an example of a delayed separation in the life cycle of the family, which may or may not conceal some pathology of the family system [48]. The process of delayed separation of men is observed in Italy and in southern Europe. On the contrary, in traditionally Protestant countries in the west and north of Europe, it occurs early around the age of 18. In southern Europe, where the sociological phenomenon of amoral familism is observed, primarily the separation of sons is delayed, with the rapid separation of daughters married off according to their parents’ plan [49]. Banfield, the creator of the concept of amoral familism, studied poor, patriarchal Catholic families in southern Italy, pointing to the association of amoral familism with conservatism and the tendency to support authoritarian, right-wing political groups. Amoral familism manifests itself in a strong conviction that honesty and moral conduct are appropriate only with the members of one’s own family, based on the principle of a sharp division between us (our own family) and “them” (strangers). In Poland, separation from the family of origin usually takes place after graduation, hence we assume that the families from the control group studied by us are not those in which the separation of the child is delayed due to the incorrect structure of the nuclear family or amoral familism. That is why in our study we selected the children’s age limit of 25 years, assuming a priori that in Poland usually young people complete their studies and move away from their parents at that age [50]. The families we studied came from a large agglomeration of one million, which in a sense excludes the influence on the delayed separation of men of amoral familism, the widespread presence of which has been demonstrated in Polish rural areas [51].

Parents of people from the control group were treated for diabetes, obesity, lipid disorders, asthma, chronic obstructive pulmonary disease, or other diseases that are dealt with in Poland by diabetologists and pneumologists. We did not analyze the distribution of the incidence of various diseases in parents of people with schizophrenia and did not compare it with the distribution in the control group, which is a certain shortcoming. On the other hand, it was the only possible way for us to recruit people from the control group, which we created by asking whether such a person lives with an adult child aged 18–25 who is still learning or studying. It is known from our research that patients with bronchial asthma and chronic obstructive pulmonary disease have some relationship with the temperament structure of the subjects [52,53]. Thus, the control group selected by us was not well matched for the representativeness of the structure of temperament in the general population. On the other hand, the severity of somatic diseases in the surveyed parents from the control group was small and usually did not interfere with their family and professional life, and in our research on temperament in bronchial asthma, we included people with a severe course and poor control of asthma. In this aspect, it can be cautiously assumed that having well-controlled somatic diseases does not significantly interfere with the functioning of the family and does not modify the structure of temperament.

It would be interesting to correlate the structure of the caregiver’s temperament with the number of years of providing care, the severity of the course of the disease in the child under care, and the number of hospitalizations and exacerbations. We did not collect such data, which is undoubtedly a shortcoming of the study. On the other hand, the young age of the charges means little difference in the years of care between caregivers. Such studies would make sense if all caregivers of patients with schizophrenia were eligible, regardless of the age of the patient. It would then be possible to determine the intragroup correlations between the variables studied. However, there would not be a good control group, because in our culture it is difficult to consider the living of an adult, single child above 25 years of age together with his/her parents as a valid cultural model of the family [54].

## 5. Conclusions

Caregivers of young adults with schizophrenia had higher values of briskness, emotional reactivity, and activity compared to the control group of parents of healthy young adults. The remaining dimensions of temperament: perseverance, sensory sensitivity, rhythmicity, and endurance did not differentiate between the two groups.

## Figures and Tables

**Table 1 ijerph-20-02119-t001:** Demographic data of caregivers of young adults with schizophrenia and control group.

Parameter	Caregivers of Young Adults with Schizophrenia (*n =* 64)	Control Group (*n* = 53)	*p* *
Age (years)	53.31 ± 10.41	53.79 ± 11.15	0.59
Number of women in the group *n* (%)	51 (80)	42 (79)	0.45
Number of men in the group *n* (%)	13 (20)	11 (21)	0.32
Maximum age of the caregiver (years)	61	59	0.42
Minimum age of the caregiver (years)	41	39	0.36

Age is given as mean ± SD. * *p*-value assessed using the Mann–Whitney U test.

**Table 2 ijerph-20-02119-t002:** Analysis of differences in temperamental traits according to the Regulatory Theory of Temperament between caregivers of young adults with schizophrenia and the control group.

Variable	Caregivers of Young Adults with Schizophrenia (*n =* 64)		Control Group (*n* = 53)		*p* *
	Mean (Stanine Score Values in Brackets)	SD	Mean (Stanine Score Values in Brackets)	SD	
Briskness	43.22 (6)	4.45	42.90 (4)	3.98	**0.032** *
Perseverance	42.50 (5)	3.29	43.10 (5)	4.37	0.392
Rhythmicity	22.46 (5)	4.15	23.01 (5)	4.27	0.348
Sensory sensitivity	42.36 (5)	3.31	43.01 (5)	3.29	0.632
Emotional reactivity	46.02 (7)	4,39	41.01 (5)	2.12	**0.012** *
Endurance	36.46 (5)	4.68	36.42 (5)	5.52	0.656
Activity	44.01.89 (7)	4.15	37.59 (5)	4.77	**0.022** *

The results along with standard deviations (SD) are presented in crude values, average values were also converted into stanine score values calculated by the authors of the test for the Polish population and given in brackets [19]. * *p*-value assessed using Student’s test. The bolded results indicate statistically significant differences.

## Data Availability

The data presented in this study are available on request from the corresponding author. The data are not publicly available due to ethical reasons.

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
