# Peer review of "The Structure of Temperament in Caregivers of Patients with Schizophrenia"

_ijerph, 2023, doi:10.3390/ijerph20032119_

Round 1

Reviewer 1 Report

The topic of the article is interesting and important, but the manuscript needs major correction.

Major comments

This section Results should be the main one in the article, but now it is the shortest and not informative.The authors are recommended to significantly improve the presentation of the results obtained. 

 In addition, the study has a lot of limitations so that this article can be published now. 

 Minor comments

Abstract:

- Lines 25-27 - specify the number of participants in the control and comparable group;

- Lines 29-31 - specify the units of measurement (points?), it is better to use p-value instead of p.

Keywords:

- Line 37 - add words that do not duplicate the words from the title of the article (this is important to increase the visibility of the article when searching in databases).

Materials and methods:

- Structure the information in the section, for example: 2.1. Date Collection; 2.2. Study Population; 2.3. Procedures; 2.4. Statistical Analysis.

- In subsection 2.1. explain how you calculated the sample size. Clearly formulate the inclusion and exclusion criteria first in the control group, then in the comparable group.

- Move Table 1 to subsection 2.2. Population study; move the note from this table under the table. Note: the age difference not significant statistically p-value = 0.59. In the Parameters column, add units of measurement for each parameter in parentheses. For example, Minimum Age (years); Maximum Age (years); Mean Age (years, Mean ±SD); Group Size (number, %); Female (number, %); Male (number, %).

- In subsection 2.3.: briefly describe the questionnaire used and the general approach to interpreting the survey results.

Results:

The name of Table 2 needs correction. Move the note under the table. Note: The results along with standard deviations are presented in crude values, average values were also....

Discussion:

Design Section 5. Limitation as an independent section of the article. Delete the numbering of paragraphs inside this section.

Conclusion:

Delete the numbering of paragraphs inside this section.

Author Response

Thank the Reviewer’s very much for your time and valuable comments on our manuscript. The responses for all points are below. The changes were introduced into the text of manuscript, as suggested by the Reviewer.

Major points:

Point 1: This section Results should be the main one in the article, but now it is the shortest and not informative. The authors are recommended to significantly improve the presentation of the results obtained.

Response 1: Thank the Reviewer’s suggestion. According to it, we made the changes in the manuscript.

Minor points:

Point 1: Abstract:

  • lines 25-27 - specify the number of participants in the control and comparable group;
  • lines 29-31 - specify the units of measurement (points?), it is better to use p-value instead of p.

Response 1:. Thank the Reviewer’s suggestion. According to it, we made the changes in the manuscript.

Point 2: Keywords: line 37 - add words that do not duplicate the words from the title of the article (this is important to increase the visibility of the article when searching in databases).

Response 2:  Thank the Reviewer’s suggestion. According to it, we made the changes in the manuscript.

Point 3: Materials and methods: Structure the information in the section, for example: 2.1. Date Collection; 2.2. Study Population; 2.3. Procedures; 2.4. Statistical Analysis.

Response 3: Thank the Reviewer’s suggestion. According to it, we made the changes in the manuscript.

Point 4: Materials and methods: In subsection 2.1. explain how you calculated the sample size. Clearly formulate the inclusion and exclusion criteria first in the control group, then in the comparable group.

Response 4: Thank the Reviewer’s suggestion. According to it, we made the changes in the manuscript.

Point 5: Materials and methods: Move Table 1 to subsection 2.2. Population study; move the note from this table under the table. Note: the age difference not significant statistically p-value = 0.59. In the Parameters column, add units of measurement for each parameter in parentheses. For example, Minimum Age (years); Maximum Age (years); Mean Age (years, Mean ±SD); Group Size (number, %); Female (number, %); Male (number, %).

Response 5: Thank the Reviewer’s suggestion. According to it, we made the changes in the manuscript.

Point 6: Materials and methods: In subsection 2.3.: briefly describe the questionnaire used and the general approach to interpreting the survey results

Response 6: Thank the Reviewer’s suggestion. Thank you very much for this comment. According to it, we made the changes in the manuscript.

Point 7: Results: The name of Table 2 needs correction. Move the note under the table. Note: The results along with standard deviations are presented in crude values, average values were also....

Response 7: Thank the Reviewer’s suggestion. According to it, we made the changes in the manuscript.

Point 8: Discussion: Design Section 5. Limitation as an independent section of the article. Delete the numbering of paragraphs inside this section.

Response 8: Thank the Reviewer’s suggestion. We removed it accordingly.

Point 9: Conclusion: Delete the numbering of paragraphs inside this section.

Response 9: Thank the Reviewer’s suggestion. We removed it accordingly.

Reviewer 2 Report

Thank you for the opportunity to review this paper and thank you to the authors for thier effort in trying to investigate an interesting and overlooked issue. 

Unfortunately, I think the paper has some major shortcomings to be considered

Major Issues

Introduction: Introduction does not present clearly the rationale of the paper as well as how the authors make hypothesis for their research. The aims of the paper are not clear.

Method: The authors, in the abstract, claim that they aim at investigate the role of temperament in influencing the choice of family members of people diagnosed with schizophrenia to take on the role of caregiver. However, to test this hypothesis, they chose parents of non-clinical people as control group. This is methodologically inadequate to verify their hypothesis. They should have chosen family members of people diagnosed with schizophrenia who did not choose to take on the role of caregiver.

Another relevant issue, in the Method, is the following: in the Introduction the authors highlight that mothers are generally more likely to take on the role of caregiver. However, in the analysis, the authors do not consider the type of position (mother, father, sister, brother), in its interaction with temperament, as well as other possible confounding variables. It could be very relevant, indeed. 

Given these relevant shortcomings, I did not further reviewed the results and discussion section.

Minor Issues.

Line 58-59: please provide reference

Line 60-63: The example doesn’t fit so well. The authors are speculating about the temperament as a possible determinant for a family member taking on the role of caregiver. The example refers to the association between burden and neuroticism. It is well known that personality constructs, especially neuroticism, are related to burden and perceived stress. But the issue, in this Introduction, is about the choice to take on a specific responsibility. This example is not good for strenghthening the core concept of the paper in the Introduction.

Line 78-80: this is a little bit confusing. The authors refer to a work they have not talked about yet.

Author Response

Thank the Reviewer’s very much for your time and valuable comments on our manuscript. The responses for all points are below. The changes were introduced into the text of manuscript, as suggested by the Reviewer.

Major points:

Point 1: Introduction: Introduction does not present clearly the rationale of the paper as well as how the authors make hypothesis for their research. The aims of the paper are not clear.

Response 1: Thank the Reviewer’s suggestion. We have made changes to the manuscript in relation to it.

Point 2: Method: The authors, in the abstract, claim that they aim at investigate the role of temperament in influencing the choice of family members of people diagnosed with schizophrenia to take on the role of caregiver. However, to test this hypothesis, they chose parents of non-clinical people as control group. This is methodologically inadequate to verify their hypothesis. They should have chosen family members of people diagnosed with schizophrenia who did not choose to take on the role of caregiver.

Response 2: Thank the Reviewer’s suggestion. We have made changes to the manuscript in relation to it.

Point 3: Another relevant issue, in the Method, is the following: in the Introduction the authors highlight that mothers are generally more likely to take on the role of caregiver. However, in the analysis, the authors do not consider the type of position (mother, father, sister, brother), in its interaction with temperament, as well as other possible confounding variables. It could be very relevant, indeed. 

Response 3: Thank the Reviewer’s suggestion. We have made changes to the manuscript in relation to it.

Minor points:

Point 1: Line 58-59: please provide reference.

Response 1: Thank the Reviewer’s suggestion. We have added reference number 15.

Point 2: Line 60-63: The example doesn’t fit so well. The authors are speculating about the temperament as a possible determinant for a family member taking on the role of caregiver. The example refers to the association between burden and neuroticism. It is well known that personality constructs, especially neuroticism, are related to burden and perceived stress. But the issue, in this Introduction, is about the choice to take on a specific responsibility. This example is not good for strenghthening the core concept of the paper in the Introduction.

Response 2: Thank the Reviewer’s suggestion. As suggested, we removed this example.

Point 3: Line 78-80: this is a little bit confusing. The authors refer to a work they have not talked about yet.

Response 3: We are very sorry, but we do not understand this remark.

Reviewer 3 Report

The authors should consider more positive language such as “living” instead of suffering.

Lines 78-81 seem more suitable for the discussion.

Lines 82-99 also seem out of place. It is here where one would expect the gap in the literature to be concisely stated, leading into the aim and methods. This paragraph could be incorporated earlier in the introduction or would not be out of place in a Table.

More information is required to detail recruitment such as brief reasons for declining to participate.

Please add 2-4 lines describing and justifying the use of the FCB-TI.

Table 1 demographic information are a result rather than method.

The limitations section, while well-written, is unusually long, and should be more concise for the reader. Indeed some of these points could be incorporated into the discussion or possible addition of a future directions section.

I find it unusual that the limitations section is numbered, but can overlook this. However, the conclusion section should be in paragraph form.

Aside from these points this is a well-written and interesting manuscript. The authors clearly have control and expertise on the topic.

Author Response

Thank the Reviewer’s very much for your time and valuable comments on our manuscript. The responses for all points are below. The changes were introduced into the text of manuscript, as suggested by the Reviewer.

Minor points:

Point 1: The authors should consider more positive language such as “living” instead of suffering.

Response 1: Thank the Reviewer’s suggestion. Thank you very much for this comment. According to it, we made the changes in the manuscript.

Point 2: Lines 78-81 seem more suitable for the discussion

Response 2: Thank the Reviewer’s suggestion. We made a minor correction in the manuscript.

Point 3: Lines 82-99 also seem out of place. It is here where one would expect the gap in the literature to be concisely stated, leading into the aim and methods. This paragraph could be incorporated earlier in the introduction or would not be out of place in a Table.

Response 3: Thank the Reviewer’s suggestion. We made a minor correction in the manuscript.

Point 4: More information is required to detail recruitment such as brief reasons for declining to participate.

Response 4: Thank the Reviewer’s suggestion. These reasons are given in the manuscript (Study population line 133-136)

Point 5: Please add 2-4 lines describing and justifying the use of the FCB-TI.

Response 5: Thank the Reviewer’s suggestion. We have described FCB-TI as suggested (line 151-156)

Point 6: Table 1 demographic information are a result rather than method.

Response 6: Thank the Reviewer’s suggestion. Thank you very much for this comment. According to it, we made the changes in the manuscript.

Point 7: The limitations section, while well-written, is unusually long, and should be more concise for the reader. Indeed some of these points could be incorporated into the discussion or possible addition of a future directions section.

Response 7: Thank the Reviewer’s suggestion. The limitations of our study are part of a discussion that is still open and therefore we wanted to highlight them in this section.

Point 8: I find it unusual that the limitations section is numbered, but can overlook this. However, the conclusion section should be in paragraph form.

Response 8: Thank the Reviewer’s suggestion. According to it, we made the changes in the manuscript.

Round 2

Reviewer 1 Report

The authors modified the manuscript. They have taken into account the main comments.

It is probably necessary to remove the dots at the end of the title and subsection headings, as well as add subsection numbers in Materials and Methods (for example, 2.1. Study population). However, this is a small technical correction.

In general, I think that the manuscript can be accepted for publication.

Author Response

Dear Reviewer.
Thank you very much for your time and such a favorable opinion about our work. As suggested, we have removed the dots at the end of the title, chapters and subchapters.

Reviewer 2 Report

Major issues

Unfortunately, the changes made by authors did not fix the issues raised during revision.

I think the biggest limitation of this paper is in methodology:

The authors’ aim is to investigate the role of temperament in influencing relatives’ choice to take on caregiving with young people with schizophrenia. However, to this aim, they chose, as a control group relatives of people without schizophrenia, instead of relatives of people with schizophrenia that did not decide to take on the role of caregiver. Temperament, for definition, is known to be a quite stable trait.

Using temperament as variable, the expected result is to use temperament as independent variable and choice to be caregiver as a dependent variable. In this case, the control group should be another one.

In the discussion section, then, the authors talk about the influence of caregiving on temperament. I think this is the most likely interpretation of results, however, if this is the case, temperament would not be an appropriate variable, that, in this case, would be dependent variable.

In my opinion, the paper with this organization is not suitable for a publication.

Minor issues

Abstract: the abstract should be not more than 200 words

Line 95-97: as the reader has not knowledge of results discussed below, I suggest not to put this consideration or at least this remark in here.

Line 116-117: what means “deliberately selected”? I think this aspect could be a limitation in the recruiting of the sample, as it makes a further possible underlying variable influencing the differences between the two groups. I would have found more appropriate to select the same way that experimental group and, in case, to use statistic to compare the groups.

Line 188-190: “The direct caregivers of patients are usually women - their mothers. As shown in our research, fathers rarely take a dominant role in the care.” The authors should provide a reference for the first sentence. Moreover, they state that their research has shown that fathers rarely take a dominant role in the care. I don’t see this finding in the results section. Where is it shown?

Line 232-233: “The other dimensions of temperament did not differ between caregivers of people with schizophrenia and the control group.” This aspect also could be worth discussing.

Author Response

Thank the Reviewer’s very much for your time and valuable comments on our manuscript. The responses for all points are below. The changes were introduced into the text of manuscript, as suggested by the Reviewer.

Major points:

Point 1: I think the biggest limitation of this paper is in methodology: The authors’ aim is to investigate the role of temperament in influencing relatives’ choice to take on caregiving with young people with schizophrenia. However, to this aim, they chose, as a control group relatives of people without schizophrenia, instead of relatives of people with schizophrenia that did not decide to take on the role of caregiver. Temperament, for definition, is known to be a quite stable trait. Using temperament as variable, the expected result is to use temperament as independent variable and choice to be caregiver as a dependent variable. In this case, the control group should be another one.In the discussion section, then, the authors talk about the influence of caregiving on temperament. I think this is the most likely interpretation of results, however, if this is the case, temperament would not be an appropriate variable, that, in this case, would be dependent variable.

Response 1: Indeed, at work there should be a control group of relatives of people with schizophrenia who have not decided to take on the role of caregiver. Unfortunately, in our opinion, it is impossible to gather such a group of doctors working with schizophrenic patients on a daily basis, because such people do not report to mental health clinics. People who do not take care do not come with a sick person to the doctor for check-ups. Temperament is indeed a relative personality trait, but under the influence of external factors in the long term it can be modified.

Minor points:

Point 1: Abstract: the abstract should be not more than 200 words

Response 1: Thank the Reviewer’s suggestion. We had to improve the abstract as suggested by the other reviewers. As suggested by the reviewer, we shortened it.

Point 2: Line 95-97: as the reader has not knowledge of results discussed below, I suggest not to put this consideration or at least this remark in here.

Response 2: We are very sorry but we do not understand this Reviewer's comment.

Point 3: Line 116-117: what means “deliberately selected”? I think this aspect could be a limitation in the recruiting of the sample, as it makes a further possible underlying variable influencing the differences between the two groups. I would have found more appropriate to select the same way that experimental group and, in case, to use statistic to compare the groups

Response 3: Thank you very much for this attention. We have already described the selection of the control group above. We just want to add that we made the selection so that the groups did not differ in terms of age.

Point 4: Line 188-190: “The direct caregivers of patients are usually women - their mothers. As shown in our research, fathers rarely take a dominant role in the care.” The authors should provide a reference for the first sentence. Moreover, they state that their research has shown that fathers rarely take a dominant role in the care. I don’t see this finding in the results section. Where is it shown?

Response 4: This is shown in Table 1 and we have made a change to the manuscript.

Point 5: Line 232-233: “The other dimensions of temperament did not differ between caregivers of people with schizophrenia and the control group.” This aspect also could be worth discussing.

Response 5: We included the information in the manuscript: Our research indirectly shows that only those variables that differentiate both groups have an impact on taking up the role of a caregiver. Other temperamental variables probably do not affect the willingness to care for sick relatives.